



*Review*
**Anthropogenically breaking macro-ecospatial 'chains'? – case review of HU Line**
Running Head: Human breaking macro-ecospatial 'chains'?
Yi Lin[1, *], Martin Herold[2]
[1] School of Earth and Space Sciences, Peking University, Beijing 100871, China
[2] Laboratory of Geo-Information Science and Remote Sensing, Wageningen University, 6708 PB
Wageningen, the Netherlands
[*] Correspondence, Email: yi.lin@pku.edu.cn
**Abstract**
Understanding of human-nature interactions is critical for global sustainability, but one of its
frontier branches, regarding intentionally-positive anthropogenic feedbacks to environment at
the macroecosystem scale, has been less studied. A concrete open question is whether people
can break those 'chain'-like macro-ecospatial transition zones. Based on remote sensing data
and integrative data analysis, we examined this issue in the case of China, which both owns a
macro-ecospatial transition zone top-ranked in the world – HU Line and has made massive
environmental restoration efforts such as the "Grain for Green" Program (GGP). Literature
reviews of the causes of HU Line revealed its natural formation, and spatiotemporal tests of
its statuses indicated its contemporary stability, both telling the inherent difficulty of shaking
macro-ecospatial 'chains'. What's worse, the limited durations of those GGP-kind endeavors
led to a debate on whether human will eventually exert positive or negative eco-effect on the
evolution of HU Line. To handle this gap, we proposed using biogeographic, bioclimatology,
and Earth system models in a simulation way and overviewed their potentials of reflecting the



complex internal, external, and integral eco-functions in human deliberately improving nature.
In all, the conclusion and proposal of this work are of fundamental implications for projecting
the future of macro-ecospatial 'chains' and pre-making polices for anthropogenically coping
with global changes in land, environment, biology, ecology, and sustainability.
**Keywords**: Macro-ecospatial transition zone, anthropogenic eco-effect, HU Line, integrative
data analysis, remote sensing data, Earth model, human-nature interaction.
**1 Introduction**
Understanding of human-nature interactions is increasingly highlighted for advancing global
sustainability (Liu et al., 2007a), and researches on the mechanisms of nature driving human
distributions and human intervening nature evolutions have massively emerged (e.g., Foley et
al., 2007; Rockström et al., 2009). However, one of its critical frontier branches, regarding the
intentionally-positive anthropogenic feedbacks to environment at the macroecosystem (those
ecosystems at the regional to continental scales) (Heffernan et al., 2014) scales and how such
ecological effects (eco-effects) work, has been less studied. A representative open question of
extensive interest is whether people can better the macroecosystem-related ecological spatial
(macro-ecospatial) layouts (Yarrow and Martin, 2007). Studies on this topic, no doubt, will be
of great implications for both more comprehensively undermining Earth's past – involving the
human-caused global changes (Crumley, 1993) and more accurately projecting Earth's future
– involving the human-accelerated global changes (Wasson et al., 2013).
Exploitation of this problem can be specified as answering a more concrete question – can
people break macro-ecospatial 'chains', i.e., regional- to continental-scale terrestrial transition
zones? This analogy is rooted in that in ecological functional regionalization (Morrone, 2006)



such a critical transition zone tends to behave as the steep environmental gradient between
two large-scale ecosystems, each demonstrating the relatively-consistent ecological condition
(Risser, 1995). The species dwelling in such a zone live near the edges of their tolerance, and
this renders the zone, in a whole sense, to be sensitive to environmental changes (Peters et al.,
2006), of course, including the potential ones due to human activities. That is, the eco-effects
of human intervening nature can be learnt via detecting the shifts of such transition zones.
The representative macro-ecospatial 'chains' include the Southern American Transition Zone
(Morrone, 2006), the Northern Australian Tropical Transect Zone (Ma et al., 2013), and the
Mexican Transition Zone (Alvarado et al., 2014), which all are underlined in the communities
of global change in environment, biology, and ecology (Heffernan et al., 2014). However, for
these transition zones few massive human improvement measures have been undertaken, and
thereby, it is inappropriate to take them into account for seeking the answer to the question.
Based on remote sensing data and integrative data analysis (Curran and Hussong, 2009),
this study tried to examine the question in the case of China. That is, as a typical large-scale
region for analyzing the influences from and the feedbacks to instable Earth's factors such as
increasing abnormal climatic events and strengthened environmental deteriorations (Liu and
Diamond, 2005; Li et al., 2016), China satisfies the two premises of seeking the answer to the
question – both occupying the macro-ecospatial 'chains' and exerting massive environmental
restorations. First, China has a distinctive macro-ecospatial layout – southeast and northwest
with distinct ecospatial properties, as divided by an imaginary demarcation line proposed by
the Chinese population geographer Hu Huan-Yong (Hu, 1935) (henceforth termed HU Line).
Initially proposed as "a geo-demographic demarcation line over China", HU Line has proved



to be able to draw the macro-ecospatial patterns of many other kinds of geo-factors in China
(Zhang et al., 2011; Liu et al., 2015). Agricultural productivity is such a typical factor (see
Supplementary Figure S1), and HU Line once locked the spatial layout of China's agricultural
economy for a long time (Wang et al., 1996). After the industrialization in China, HU Line
has still kept locking the macro-spatial layout of China's economic developments, which
depended on the earlier-stage accumulations from agriculture economy and other supportive
geo-factors, e.g., water resource (Fang et al., 2015). So far, HU Line is still working (Gaughan
et al., 2016) (Figure 1). Second, China has made massive environmental restoration efforts
such as the "Grain for Green" Program (GGP) and the "Three-North Belt" Program (Liu et al.,
2008) (Figure 1) and now shows the largest greening area all over the world (State Forestry
Administration of China, 2009). These efforts can characterize human's intentionally-positive
macro-scale feedbacks to land surface developments. Hence, HU Line can be assumed as the
optimal case for probing the eco-effects of the intentionally-positive anthropogenic feedbacks
to environment at the macroecosystem scale.

-Insert Figure 1 here-

The following was dedicated to seeking the answer to the question of "whether can people
break through macro-ecospatial 'chains'?", via a literature review of the HU Line-associated
previous studies and a comprehensive overview of their inferences.
**2 The macro-ecospatial 'chain' – HU Line: causes and statuses**
To obtain the answer to the question, the first step is to examine the causes and statuses of
this macro-ecospatial transition zone, and this facilitates proposing the essence-oriented and
reality-rooted plans to break through it. First, the formation of HU Line proved to be decided



by a diversity of geo-factors across China (Hu, 1935). Large terrain elevation gradients affect
atmospheric circulations, leading to spatially-varying precipitation richness (Zheng and Liou,
1986); heterogeneous temperature accumulations alter the growth phases of plants, resulting
in regionally-different crop productivity and food supplies (Hou et al., 2014); some kinds of
ecosystems are preferred by life, as they can provide more livable conditions for the species
(Millennium Ecosystem Assessment, 2005); disasters tend to result in the transfers of livable
residential areas (Li et al., 2014). A systematic analysis of the specific eco-effects caused by
such geo-factors, which should play the dominant roles in causing and maintaining HU Line,
was expanded by using the approach of integrative data analysis (Curran and Hussong, 2009;
Chand et al., 2017) as follows.

-Insert Figure 2 here-

2.1 Natural causes
*Topography.* China's landform looks like "a three-step ladder lowering from west to east"
(Figure 2a), and this topography serves as the basis of forming the spatial patterns of natural
and human geography in China (Xu et al., 2015b). The highest Tibet Plateau on the Earth is
the first-step ladder (on average ~4, 000 m, termed Third Pole) (Qiu, 2008); the second-step
ladder is briefly composed by the Yunnan-Guizhou Plateau, the Loess Plateau, and the Inner
Mongolia Plateau, with their average altitudes at 1, 000–2, 000 m; the third-step consists of
the Northeast Plain, the North China Plain, and the Yangtze River Basin, with their altitudes
below 500 m. The first-step ladder – the Tibet Plateau – as a dynamic attractor increases the
thermal contrast between the relief and the Pacific ocean, and this influences the East Asian
summer monsoons in terms of moist static energy (Chen and Bordoni, 2014); the layout of the



whole ladder is primarily in the northeast-southwest direction, which shifts the rain band and
moisture transport modes in the East Asian monsoon climatic system (Xu et al., 2015b). That
is, the topography of China breaks the common pattern of climatic distributions that exhibits
gradients in latitude, and acts as the prime determinant of HU Line formation.
*Precipitation.* The formation of HU Line is partially regulated by the spatial variability of
precipitation over China. This point is remarkably reflected by the large section of HU Line in
southwest China, which lies in the first-step ladder instead of between any two steps (Xu et al.,
2015b). This "breakthrough" at that local region was rooted in the availability of rich rainfalls
there, as evidenced by the spatial distribution of annually average precipitation (Figure 2b).
The terrains with rich precipitation of 1, 600–1, 800 mm are dominantly located in southeast
China, while scare precipitation (<100 mm) primarily in northwest China. Compared to the
northwest–southeast direction of topographic lowering, the spatial distribution of precipitation
is more approximate to the common one of climatic factors changing from north to south (Xu
et al., 2015b). Further, precipitation corresponds to water resource, which is closely related to
agricultural productivity (Chen et al., 2013); in addition, the climatic factor of precipitation
helps to determine the spatial pattern of population distributions as well (Piao et al., 2010). In
all, the macro-scale eco-effect of precipitation is both pushing the southwest end of HU Line
beyond the topography-decided layout (Figure 2a) and strengthening HU Line's formation.
*Temperature.* The climatic factor – temperature – also plays a vital role in regulating the
spatial layout of HU Line, since temperature is able to change the spatial modes of ecosystem
yields thru adapting their phenology (Tao et al., 2006; Chen and Xu, 2012). For the southwest
end of HU Line (Figure 2c), temperature performs with a similar function as precipitation, i.e.,





adequate precipitation and warm temperature together favoring agricultural production and
human regeneration. For the central part of HU Line, temperature also helps to break through
the eco-effect of the topographic gradient between the second- and third-step ladders, namely,
although precipitation is in short, temperature can somehow favor agricultural activity (Chen
et al., 2013). The northeast end of HU Line also deviates from the step-jumping line of the
ladder but is conversely located in the third-step ladder. The reason is that although the related
plain terrains are propitious to agricultural productions the inverse conditions about rainfalls
and, particularly, temperature drags the demarcation line to the south. Based on the analyses
of topography, precipitation, and temperature, it can be inferred that the formation of HU Line
is a result of synthesizing the eco-effects of multiple such geo-factors.
*Agricultural productivity potential.* The integral eco-effect of the primary geo-factors such
as topography, precipitation, and temperature deciding HU Line can be pictured by deriving
the index of cropland Potential Productivity of Radiation and Temperature (PPRT) (Yang et
al., 2010) (Figure 2d). As a result of solar radiative energy, terrain, rainfall, and temperature
co-functioning on crops, this typical indicator can play a key and far-reaching role in driving
agricultural development and modernization and forming the basis of people residence (Yang
et al., 2010). The PPRT in the southeast China is much higher than those in the other regions,
and the lowest PPRT lies in the northwest (Figure 2d). The PPRT map can preliminarily draw
the shape of HU Line, although in its central section there are a couple of "bulges" sticking
into the northwest. The contrast of agricultural productivity potentials across HU Line is so
distinctive that both how these geo-factors integrally can render the formation of HU Line and
why HU Line locks the macro-ecospatial layout of China can be intuitively understood.



*Soil erosion.* In addition to the geo-factors integrally deciding the prime macro-ecospatial
layouts of China as analyzed above, some other geo-factors with inverse eco-effects may be
able to reshape HU Line. A typical case is soil erosion, which can directly impact agricultural
production and people living. The types of soil erosions are diverse, and the mainstream ones
over China include water erosion, wind erosion, and freeze-thaw erosion (www.moa.gov.cn).
Soil loss through water erosion is the most serious land degradation in China. The annual soil
loss triggered by water erosion reaches about 5 billion tons. In north China, land degradation
is seriously caused by wind erosion, which covers ~379, 600 km$^2$ and is mainly distributed in
the arid and semi-arid regions where the total annual rainfall is below 500 mm. After water
erosion and wind erosion, freeze-thaw erosion is the third most serious soil erosion type. Jin
et al., (2015) found that in the Qinghai-Tibetan Plateau the probability of soil freeze is above
80%, but in south China (below 35°N) the probability is lower than 20%, which, instead, is
caused by the exceptionally cold current. In north China (above 35°N), the probability of soil
freezing is 30%–50%, while in the northeast China, it is about 40%–80%. The northwest side
of HU Line, hence, can be referred to as a region of macroecosystem-sense land degradation.
The map of overlapping the spatial distributions of the three kinds of soil erosions in terms of
erosion intensity is displayed in Figure 2e. It can be realized that the two regions circled by
the two "bulges" sticking into the northwest (see Figure 2d) are overlapped by soil erosion,
and this inverse effect pushes the PPRT-drawn demarcation line in central China moving
southward, approaching the defined HU Line (Hu, 1935).
*Large-scale disasters.* In contrast to soil erosions that play their roles slowly in a long run,
natural disasters work in a relatively quicker but more destructive way, particularly for those





occurring at the large scales. The typical cases of large-scale disasters in the northwest China
include desertification, freeze-thaw, and avalanche, whose distributions are listed in Figure 2f.
In fact, through the statistics of the frequencies of such dominant natural disasters over China
in the history, Liu and Yang (2012) found that the occurrences of such disasters for different
types were common and their spatial distributions were significantly different from each other.
Guan et al. (2015) reported that geographically, the occurrences of frost and snowstorm were
more frequent in the northwest China. Wang et al. (2004) found that in the arid and semi-arid
zones of fragile eco-environment in north China, numerous areas have been facing the danger
of severe sandy desertification. Jin et al. (2015) noticed that freeze disasters mostly happened
in northwest China, and so did avalanches (Wang and Huang, 1986). In combination with soil
erosion, such large-scale natural disasters have kept truncating the "bulges" (Figure 2d) of the
topography-precipitation-temperature-deciding demarcation line, eventually forming HU Line
– the prime macro-ecospatial 'chain' over China.
As the essence-oriented analyses exerted above, the primary Earth processes leading to the
formation of HU Line have been figured out. The formation ranged from topography drawing
its skeleton, precipitation, temperature and agricultural productivity potential straightening its
sections, to soil erosion and large-scale natural disasters truncating its "bulges". In summary,
the overview of the causes of HU Line suggested the inherent difficulty of breaking any
macro-ecospatial 'chain'. After all, it is hard for people to practically interfere those powerful
natural geo-factors at the macroecosystem scale.
2.2 Spatial and temporal statuses
Exploring the aimed question also needs to consider the contemporary statuses of HU Line.



This is due to that HU Line was noticed in last century (Hu, 1935), and the concerns about its
situations unavoidably arise – has HU Line altered when compared to 1935? To better grasp
the current statuses of this macro-ecospatial 'chain', the questions about whether HU Line so
far remains stable, specifically, spatially without new local "bulges" emerging and temporally
without its whole location shifting for decades, were explored via macro-ecosystem analysis,
often based on remote sensing data capable of covering large areas (Heffernan et al., 2014).

-Insert Figure 3 here-

*Spatial stability*. The key point of investigating the spatial variability of HU Line is to first
propose proper criteria that can characterize the changes of this critical terrestrial transition
zone. As analyzed above, HU Line, in effect, is a result of fusing several geo-factor-related
demarcation lines, which do not coincide. This, in turn, means that exploration of its spatial
stability is equivalent to evaluating its covered natural land surfaces, and this involves a high
complexity of environmental indices (UNEP, 2002). This complexity can be figured out by
referring to the indices proposed and evaluated in the Global Environment Outlook Program
(UNEP, 2002; 2007), the Heinz Center Evaluation of America's Ecosystems (the Heinz Center,
2002), the Millennium ecosystem assessments (Millennium Ecosystem Assessment, 2003;
2005), and the Ecological Indicators for The Nation Proposed by The U.S. National Research
Council (NRC, 2000). The situation is approximately complicated when regarding China, as
illustrated in the study of the natural environment of China based on topography, temperature,
water, biology, soil, and other geo-factors (Yang et al., 2002). After comparing several typical
ones of such geo-factors, Gao et al. (1999) concluded that the most key geo-factors related to
human life are hydrothermal conditions, and Wang et al. (2008) discovered that temperature,





precipitation and sunlight conditions are the kernel factors deciding agricultural productivity.
Yang and Ma (2009) further derived nine geo-factors for evaluating the natural environmental
suitability and derived that the eco-effects of all of these nine environmental indices can be
integrally reflected in terms of ecosystem distribution pattern. The spatial layout of the major
terrestrial ecosystems across China is shown in Figure 3a. It can be qualitatively interpreted
that HU Line now still spatially marks the prime macro-ecospatial transition over China.
*Temporal stability.* A typical geo-feature often used for reflecting the temporal stability of
HU Line is vegetation growth status, which is very sensitive to environmental changes and
also can be readily characterized using the remote sensing retrieved parameter of Normalized
Difference Vegetation Index (NDVI) (De Keersmaecher et al., 2014). Specifically, this study
derived the spatial pattern of the interannual variations of the NDVIs at their growing seasons
from 1981 to 2011 (Figure 3b), in terms of the parameter of NDVI fluctuation intensity that in
this work was defined as the variance of the maximum NDVI values extracted per five years.
The used NDVI dataset was the Global Inventory Modeling and Mapping Studies NDVI3g
data (https://nex.nasa.gov/nex/projects/1349) (Zhu et al., 2013). It was noticed that the areas
with strong variations are briefly located along HU Line. No matter the northwest regime of
low vegetation densities or the southeast with dense vegetation covers both show relatively
weaker interannual NDVI variations. This result suggested that HU Line is comprised by a
series of short-term (annual-scale) unstable ecotones (Wasson et al., 2013), which are easily
affected by interannual environmental and climatic oscillations; on the other hand, this also
illustrates the long-term (decadal-scale) stability of this prime macro-ecospatial 'chain' in
China.





Overall, the analysis of the status of HU Line suggested the inherent difficulty of breaking
macro-ecospatial 'chains'. After all, increasingly severe global changes (Wasson et al., 2013)
still could not alter the statuses of those critical transition zones at the macroecosystem scales,
let alone human endeavors.
**3 Breaking the macro-ecospatial 'chain' – HU Line?**
The overview of the causes and statuses of HU Line clarified the difficulty of breaking this
'chain', but did not exclude the possibility. From a scientific perspective, the potential ways
of actively restoring environment such as the GGP-kind activities (Liu et al., 2008) shall be
explored for the next generations. This tells the challenge but significance of launching this
study. Consequently, theoretical analyses about the modifiability of HU Line and the diversity
of anthropogenic eco-effects on HU Line were carried out in order to comprehensively derive
an answer to the question.
3.1 Theoretical analysis of HU Line modifiability
Although the analyses on the statuses of HU Line presented its current spatial and temporal
stability, this cannot assure its steadiness in future. The reason is that the currently-stable HU
Line actually is bearing more and more complex eco-pressures (Rain et al., 2007) than ever. A
representative case is that the percentage of population at the eastern side of HU Line dropped
from 96.0% in 1935 down to 93.8% in 2000, while at the western side rose from 4.0% up to
6.2% (Qi et al., 2016). It seems that this percentage-indicated change of population is minor,
but the number of population change at the western side of HU Line – an increase of almost
20 million from 1935 to 2000 (Wang et al., 2008) – is quite large due to the giant population
base of China. This massive population increase at the western side of HU Line can illustrate





the complexity of eco-pressures (Rain et al., 2007). That is, population increases have proved
to impose more eco-pressures to the natural environments at the western side of HU Line, but
this factor, on the other hand, serves as an essential positive strength for putting the GGP-kind
Programs into practice (Liu et al., 2008).
Theoretically, such two counter aspects of eco-effecting by the same factor reflect both the
possibility of anthropogenically breaking HU Line and its challenging, i.e., the eco-balancing
between human and nature can be broken but will always be a hard work (Foley et al., 2007;
Rockström et al., 2009; Alvarado et al, 2011), particularly towards the direction of exhibiting
positive eco-effects in the end. This is why the Chinese Premier's question – "can we break
through the 'lock' of HU Line?" was posed and then quickly sank into a debate (Chen et al.,
2016). The mainstream answer given by geographers is that the ecospatial layout of China
will not become totally different within a relatively long time, but the northwest region can
achieve higher-degree modernization and higher-quality urbanization (Chen et al., 2016). The
reasoning is based on that in the tomorrow of China's eco-societal development, the limit by
the spatial imbalance of food productions can be handled through manual deployments, but
the constraints by water resource and other inverse geo-factors will be strong yet (Wang et al.,
2012). Thereby, all of such geo-factors may render that HU Line is likely to continue to lock
the whole spatial patterns of China's biology, ecology, and development.
However, such a common-sense answer by geographers (Chen et al., 2016) is not enough
for determining the modifiability of HU Line in details, e.g., starting from which point on HU
Line to achieve the goal. After all, breaking HU Line does not require shifting the whole HU
Line. Instead, just adjusting one or two sections at local regions can also be considered as a



breakthrough to HU Line, in a similar scheme as the historic process of HU Line forming in a
way of section by section. In retrospect, the dynastic data of China's households (Liang, 1980)
told that HU Line emerged merely after about A.D. 1240. Specifically, before that time, the
spatial pattern of population distribution in China followed HU Line only in its southern end
(Wang et al., 1996); for the regions beyond 30°N, the spatial distribution of population was
distinguished along the longitudinal or altitudinal directions. Before the 13th century, there
was a higher ratio of people living in northwest China. This coincided with the distribution
characteristics of precipitation before A.D. 1240 (Zhang and Crowley, 1989). Then, more and
more people migrated to the southeast of China. This massive migration was driven by the
climatic changes during the following period, as evidenced by both the records of China's
historical climates (Zhang, 2005) and the results of model-based historic climate simulations
over China (Man et al., 2012). China had experienced twice massive abrupt climatic changes
at about A.D. 880 and 1240 (Zhang et al., 1994). That change occurring in the 13th Century
was the largest climate change event across China during the past 2000 years. The effect of
that change on the climatic mode of China lasted until today. People's excessive and random
lumber harvesting strengthened this trend and enhanced the formation of HU Line (Wang et
al., 1995). All of these processes, at least, tell the modifiability of HU Line.
3.2 Complex eco-effects of anthropogenic feedbacks
Exposing the modifiability of HU Line does not mean having a confirmative answer to the
question. The substantial reason is due to the complexity in the eco-effects of anthropogenic
feedbacks to nature. This is illustrated by an open question as people often ask – "whether the
known laws of nature development need to be obeyed or more human interventions shall be





encouraged to promote the environmental quality?".
Human's, no matter intentionally positive or unintentionally negative, feedbacks to nature
usually show complicated eco-effects, particularly from the perspective of macro-ecosystem
analysis at the large scales and in a long run. For the environmental restoration policies with
intentionally positive effects, their final effects still need to be explored further (Xiao, 2014).
The anthropogenic feedbacks to nature also include China's industrial consequences such as
economic development (Naughton, 2007) (Figure 4a), which causes resource consumptions
(Zhang et al., 2011) but, on the other hand, financially ensures the practice of environmental
restoration policies (Liu et al., 2008). These feedbacks that typically display two contrasting
aspects reflect the complexity of seeking to break macro-ecospatial 'chains'.

-Insert Figure 4-

*Intentionally-positive activities but showing uncertain eco-effects*. In fact, there have been
no absolute conclusions for most of the environmental restoration endeavors, when regarding
their final positive or negative eco-effects. The GGP-kind Programs (Liu et al., 2008) were
often used as the representative cases for studying if human interventions with intentionally
positive purposes can improve the macro-ecospatial modes of China. Xiao (2014) reexamined
the biophysical consequence of GGP on the Loess Plateau and observed that the feedbacks of
GGP to regional climates depend on the negative forcing from both carbon sequestration and
evapotranspiration and on the positive forcing from lower albedo; nevertheless, Xiao (2014)
suggested that further work is still needed to assure the net eco-effects of GGP on the regional
climates. Feng et al., (2016) found that the GGP operated on the Loess Plateau for long, now,
is approaching its sustainable water resource limits, and the future evolutions of the relevant



terrestrial ecosystems will be full of uncertainties. These studies all told that the intentionally
positive programs may show uncertain eco-effects eventually. So, there are a lot of unsolved
works ahead, regarding the complicated interactions between the restored ecosystems and the
macro-scale geography.

*Commonly-deemed negative activities but behaving with favorable eco-effects*. On the other

hand, more and more studies found that the generally-classified negative activities show some
kinds of positive eco-effects. A typical case is the factor of quickly-increasing atmospheric
nitrogen deposition ($N_{dep}$), which may degrade human health, alter the chemical compositions
of water and soil, break greenhouse gas balancing, and decrease biological diversity (Liu et al.,
2015). The spatial distributions of the averaged $N_{dep}$ between 2003 and 2014 were derived by
Kriging interpolation of the data from 41 $N_{dep}$ monitoring sites, and its dropping mode from
the southeast to northwest over China (Liu et al., 2015) was marked by HU Line (Figure 4b).
However, this inverse factor also proved to enhance forest growth and carbon sequestration
(Yu et al., 2014), and atmospheric circulations can gradually transport it far and expand its
eco-effect range beyond HU Line. Another case involves economic development. As a major
anthropogenic factor disturbing macro-ecological balances, e.g., coal mining easily leading to
land subsidence (Sahu and Lokhande, 2015), economic development often means resource
over-consumption, as illustrated by the spatial layout of annual water use density over China
(Supplementary Figure S3). Meanwhile, this factor also exemplifies the beneficial eco-effects
of anthropogenic feedbacks to nature. In fact, the achievement of economic development has
financially supported implementing environmental restoration, and this favor is achieved via
actively transferring the surplus finances from southeast to northwest by the Chinese Central



Government. However, this kind of anthropogenic feedbacks in a macro sense, particularly
for their indirect and implicit eco-effects, has been less considered in the physical studies of
human-nature interactions in Earth sciences, e.g., with just simple relevant tools in the minor
simulation models considering such eco-effects (Xiao, 2014).
Overall, although the modifiability of HU Line has been briefly verified, it was also found
that it is a challenging task to conceive the specific plans of breaking it. The reason is that the
anthropogenic feedbacks to nature are not mechanistic or linear, beyond the traditional regime
of landscape ecology (Peters et al., 2006). This suggested that more comprehensive methods
of macroecosystem ecology (Heffernan et al., 2014) capable of characterizing the underlying
internal and external eco-effects of these varying feedbacks are needed to project the future of
HU Line.
**4 Comprehensive exploration of HU Line evolution**
To achieve a comprehensive projection of the future of HU Line, a sound tactics of probing
its evolution is to introduce efficient ecosystem process simulation models, which can handle
the complexity of human-nature interactions (Liu et al., 2007b). In retrospect, the researches
following this strategy have already emerged. For instance, in order to resolve the evolution
of population geography across China during the past 2, 000 years, some simulation models
such as the agent-based models (Wu et al., 2011) have been attempted to make the analyses.
However, comprehensively projecting the future of HU Line relies on more all-around models.
After all, HU Line marks almost the whole-sense biosphere eco-mode, far beyond population,
over China. Hence, we proposed introducing different-regime simulation models to decipher
the built-in code of this macro-ecospatial 'chain', involving its internal balance, external drive,





and integral function. The potential simulation models can be classified into three types, i.e.,
biogeographic models (HU Line serves as the eco-media interlinking biology and geography),
bioclimatology models (as the eco-interface for mirroring biology's responses/feedbacks to
climate forcing), and Earth system models (as the eco-frontier interacting with all of the other
kinds of geo-factors), and their regime relationships are shown in Figure 5.

-Insert Figure 5 here-

4.1 Internal eco-effect – biogeographic model
HU Line marks the distributions of species and ecosystems over China in the geographical
space, relating to the key "distribution" branch in the domain of biogeography. Accordingly,
the future of HU Line can be projected using biogeographic models, which reflect the internal
eco-effects between biology and geography. This schematic plan has been attempted in the
previous studies. Ni et al., (2000) used an equilibrium terrestrial biosphere model BIOME3 to
yield a simulation of plant distribution, which proved to be in good agreement with the related
natural vegetation map. Later, Ni (2001) better described the vegetation distribution by further
proposing a new biome classification method that considers soil conditions, eco-physiological
parameters, and the competitions between plant functional types. Recently, macroecological
factors capable of explaining the large-scale spatial patterns of populations (Xu et al., 2015a)
have attracted more attention. Along with more such underlying bio- and geographic-relevant
factors considered, biogeographic models can better forecast the evolution of HU Line.
In order to better explore the substantially-dynamic internal eco-effects between biology
and geography, their interaction processes are increasingly concerned in the development of
biogeographic models. Tan et al., (2013) adopted such a process-based ORCHIDEE model to



397 estimate the carbon fluxes and stocks of the grasslands and derive the spatial distributions of

398 the leaf area indices of vegetation and soil organic carbons across China. Sasai et al., (2016)

399 operated the BEAMS model that integrate eco-physiological and mechanistic approaches and

400 satellite data to predict net ecosystem production over eastern Asia, and they observed that the

401 integral effects of the factors capable of controlling net ecosystem productivity are positive in

402 the southern part of East Asia but negative in the northern and central parts of East Asia. In

403 the same way, with process-based biogeographic models used, the performance of HU Line

404 developing at the finer temporal scales can be projected further.

405  Studies of biogeographic patterns have also begun to take various biogeochemical factors

406 into account. Han et al., (2011) derived the spatially-varying modes of 11 chemical elements

407 such as nitrogen, phosphorus, and potassium in the leaves of 1900 plant species over China.

408 The concentrations of these elements proved to have significant latitudinal and longitudinal

409 trends, principally decided by soil and plant functional types. Such findings (Han et al., 2011)

410 facilitate pushing forward the introduction of more biogeochemical models into the projection

411 of HU Line's evolution. In addition to environmental changes, invasion species may also alter

412 the ecospatial modes over China (Wu et al., 2010). To further co-characterize these effects,

413 more complex biogeochemical processes like soil transmitting helminthic infections (Lai et

414 al., 2013) can be modeled further. Lai et al., (2013) noticed that higher infection prevalence

415 (>20%) with the whipworm *Trichuris trichiura* occurred in a few regions of south China,

416 whereas very low prevalence (<0.1%) of hookworm and whipworm infections was mainly

417 found in north China. These simulations proved to fit the real spatial patterns of the invasion

418 species. Overall, the future evolution of HU Line can be better projected by taking its internal





eco-effects into account more, based on the conventional biogeographic models to the newest
biogeographic models with modules capable of simulating complex biogeochemical processes
incorporated.
4.2 External eco-effect – bioclimatology model

Along with the trend of exploring biogeographic models with internal eco-effects (e.g., Ni

et al., 2001) increasingly emphasized, active external drives were also highlighted. The most
representative case is climate factors such as ambient $CO_2$ and climate changes (Gao and Yu,
1998). To reflect such external eco-effects, bioclimatology models have been introduced for
directly or indirectly investigating those macro-ecospatial patterns. The direct investigations
were fulfilled through, e.g., a coupled ocean–atmosphere general circulation model to infer a
doubled greenhouse gas scenario for 2070–2099 (Ni et al., 2000). In this simulated climatic
scenario, the carbon stocks in the deduced vegetation maps may increase significantly, both
with and without the $CO_2$-related direct physiological effects. Via linking the Crop-C model
with the climate change scenario projected using a coupled FGOALS bioclimatology model,
crop net primary production across China for 2000–2050 was simulated (Zhang et al., 2007),
and it was projected that a higher increase would occur in a majority of regions in the eastern
and northwestern China. Sun and Mu (2013) proposed a new CNOP-P (conditional nonlinear
optimal perturbation related to parameter) method of to generate a possible climate scenario
and to study the influence of climate change on the simulated net primary production in China
with a LPJ-DGVM dynamic global vegetation model. It was also realized that net primary
production decreases in northern China and increases in northeast and south China when the
temperature varies as a result of a CNOP-P-typed temperature change scenario (Sun and Mu,





2013). Jing and Li (2015) reported that when regarding the eco-spatial pattern, the projected
net primary production and net ecosystem production decreases were primarily located in the
tropical and temperate regions. These studies suggested that bioclimatology models can be
used to project the dynamic eco-spatial patterns of China in the future.

The external effects of climate on macro-ecospatial modes may also be indirectly done by

changing the other biology-related environmental factors. Wang et al., (2012) run the variable
infiltration capacity model to assess the implications of climate change for water resources in
China, and their findings indicated that the annual runoff over China as a whole will probably
increase by about 3–10% by 2050 but with uneven spatial distributions. The prevailing mode
of "north dry and south wet" in China is likely to be exacerbated under global warming (Wang
et al., 2012). Chen and Frauenfeld (2014) further projected that by the end of the 21[st] Century,
temperature may increase by 1.7–5.7 ℃, with strengthened warming over northern China and
the Tibetan Plateau. Four regional climate models were used to simulate the climate of the last
20-year summers (1989-2008) across China, and both the observed and simulated linear trend
of precipitation shows a drying trend over the Yangtze River Basin and wetting in south China
(Wang et al., 2016). Based on 17 models in the Coupled Model Intercomparison Project phase
5 (CMIP5), the simulation of precipitation extremes in China was assessed under the baseline
climate condition compared to a gridded daily observation dataset CN05.1 (Guo et al., 2016).
From water resource to temperature extreme, their links to the biology in the bioclimatology
models can also help to project the possible changes of spatial patterns involving HU Line.

Although the reviewed bioclimatology models can help to give more dynamic information,

it was also realized that different models may give predictions with biases. For example, Jing





and Li (2015) indicated that the differences in the accelerating terrestrial carbon losses due to
global warming estimated by the CMIP5 models ranged from 6.0 Tg C yr$^{-2}$ in CESM-BGC to
52.7 Tg C yr$^{-2}$ in MPI-ESM-LR. Given the complexity of vegetation dynamic patterns under
global climate change, multi-scale spatiotemporal explicit models are necessitated in order to
account for land surface heterogeneity. Chen and Frauenfeld (2014) examined the changes
under three emission scenarios in the 21$^{st}$ Century on the basis of a multi-model ensemble of
20 general circulation models under the CMIP5 frame. Qiu et al. (2016) further proposed a
Multi-Scale Spatio-Temporal Modeling framework to derive, reconstruct, and test multi-scale
vegetation dynamic patterns under global climate change in China. These strives suggested
that current bioclimatology models can handle the prediction uncertainties such as the scaling
effects (Qiu et al., 2016), and hence, we proposed also using bioclimatology models for better
projecting the specific evolutions of HU Line for its different segments.
4.3 Integral eco-effect – Earth system model
Those representative biogeographic and bioclimatology models as reviewed above are still
far from enough for comprehensively characterizing all of the geo-factors that may trigger the
changes of the macro-ecospatial patterns over China. To reflect the integral effect by all of the
potential impact factors on the future evolution of HU Line, we proposed a potential strategy
of fusing multiple models, which can be heterogeneous or homogenous. As we know, parallel
with the schematic frames of biological plus geographic models and biological plus climatic
models, different land surface models could be coupled with four regional climate models to
reconstruct the summer climate (1989–2008) in China (Wang et al., 2016); the combination of
multiple climate models was attempted to reproduce the spatial distribution of precipitation





extremes over China (Guo et al., 2016); those multimedia models like POPsLTEA (Song et al.,
2016) were also developed to assess the possible impacts of climatic change on the fate and
transport of polycyclic aromatic hydrocarbons in East Asia. Overall, this model-combination
proposal can help people to better infer the future of HU Line by just grasping limited kinds
of models.

The second strategy is to expand the area of analysis beyond China, even out of terrestrial

Asia. China is annually affected by the Pacific monsoons. Wang et al., (2008) used a coupled
regional ocean-atmosphere model of P-sigma RegCMg-POM to derive the spatial patterns of
the climatic mean extreme precipitation thresholds, which were characterized by a few huge
value centers covering north and part of northeast China, Yangtze-Huaihe River valleys, and
south China. Siew et al., (2014) evaluated future climatic changes through the representative
concentration pathways, using ten coupled atmosphere-ocean general circulation models in
the CMIP5. Such models can help to deeply understand HU Line at the broader scales. At the
same time, macrosystem ecology (Heffernan et al., 2014) substantially constructed under the
hierarchical framework also highlights the eco-effects at fine scales. More functional modules
capable of exposing the underlying associations between local restoration efforts and regional
ecological patterns (Xiao, 2014) need to be developed and operated. With the broad and fine
scales interlinked, the puzzles concerning macro-ecospatial 'chains' can be more easily solved
under these scenarios.

The extreme solution plan is to use Earth system models to integrally simulate the various

effects. After all, the Earth factors potentially influencing the spatial pattern of HU Line are of
considerable diversity, ranging from underground to ionosphere. For example, groundwater



depletion impacts crop productions in the major global agricultural regions and has negative
ecological consequence (Doll et al., 2014). Doll et al., (2014) used a new version of the global
hydrological model – WaterGAP and found that India, United States, Iran, Saudi Arabia, and
China showed the highest groundwater depletion rates in the first decade of the 21st Century.
This analysis can be strengthened by enhancing the spatial resolution in China. Wang et al.,
(2010) fused the tropospheric atmosphere chemistry TACM model with the regional climate
RegCM3 model to build a new regional climate chemistry modeling system RegCCMS model
for the purpose of analyzing the spatial-temporal distributions of anthropogenic nitrate aerosol,
radiative forcing and climate effect in China. With such models, people can adapt the settings
of their parametric conditions to explore the potential influences of the intentionally-positive
anthropogenic effects, e.g., artificially adding vegetation covers close to HU Line to examine
their surrounding evolutions via the operations of the model simulations, but the inferences
may be different due to the varying data availability and temporal periods of interest. Overall,
as we proposed, with more comprehensive Earth system models used, the future evolutions of
HU Line, no doubt, can be more accurately and stably projected.
**5 Summary**
In the case of China that occupies the critical terrestrial transition zone – HU Line and has
made massive restoration endeavors, this study examined the potential of anthropogenically
breaking macro-ecospatial 'chains', ranging from their natural causes, spatiotemporal stability,
complexity of human-nature interaction in eco-effect to model simulation, through reviewing
the relevant studies and integrative data analyses. The piled-up knowledge is of implications
not only for biogeosciences on regional\global changes but also for biosphere conservation,

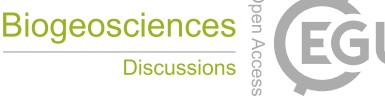

climate adaption, environment maintaining, nature protection, and macroecosystem ecology.
Finally, the proposal of using various simulation models has pointed out a way for exploring
the anthropogenic effect on critical transition zones. Yet, it must be noted that there have been
none models, even with Earth system models counted in, capable of fully characterizing the
complex processes of human-nature interactions and clearly identifying the positive\negative
eco-effects of people's, even though intentionally-positive, feedbacks to nature. This means
that the question, substantially, shall be posed to the whole community, and it is expected that
this review will inspire more studies in this direction – anthropogenic eco-effect on nature and
its macro-ecospatial pattern.

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

**Acknowledgements**
The authors would like to sincerely thank those who distributed their research results, which
have been instrumental in accomplishing this study. This study was financially supported in
part by the National Natural Science Foundation of China (Grant No. 31870531 and
31670718), and in part by the National Key Research and Development Program of China
(2017YFC0210102).

**Conflict of interest**
The authors have no competing interests.

**Authors' contributions**
Y.L. designed the study and analyzed the data. Y.L. and M.H. both contributed to paper writing.

**Data availability**
The used data are available from https://nex.nasa.gov/nex/projects/1349 and www.moa.gov.cn.









**Figure Captions**

Figure 1. Illustration of China's population distribution across HU Line and
already-implemented programs for environmental restoration. HU Line draws the distinctive
population density map of China (Census year: 2010) (We generated the image based on the
data that was published in China Statistical Yearbook, 2010). The obvious environmental
improvement from 1986 to 2016 shows the positive anthropogenic feedbacks, e.g., the "Grain
for Green" Program (GGP) (Liu et al., 2008). Note that some of the figures for illustrations in
this review did not include the data for the districts of Hong Kong, Macao, and Taiwan in
China.

Figure 2. **A**. China terrain elevation map; **B**. China average annual precipitation (AAP) map;
**C**. China annual average temperature (AAT) map; **D**. China cropland potential productivity of
radiation and temperature (PPRT) map (Census: 2000) (We generated the image based on the
data that was published in Yang et al., 2010); **E**. China soil erosion degree map (Census:
1995); **F**. China large-scale natural disaster map (We generated the images of **A**. **B**. **C**. **E**. **F**.
based on the related data that were published in www.moa.gov.cn).

Figure 3. **A**. China's dominant terrestrial ecosystem map (Data from www.moa.gov.cn); **B**.
China NDVI fluctuation intensity map (1981-2011) (We generated the images based on the
NDVI3g data that were published in Zhu et al., 2013).





Figure 4. **A**. China gross domestic productivity map (Census: 2010) (We generated the image
based on the data published in China Statistical Yearbook 2010); **B**. China inorganic nitrogen
wet deposition map (We generated the image based on the data that was published in Liu et
al., 2015).

Figure 5. Schematic framework of integrating the simulation models from different regimes
for exploring the possible evolution of HU Line. The different-regime models together can
better characterize the complicated interactions that physically, chemically, or physiologically
lead to a lot of cross-regime eco-effects capable of directly or indirectly influencing
environment. Such eco-effects cover black-carbon lowering snow surface albedo (Hadley and
Kirchstetter, 2012), eutrophication deteriorating acidification of subsurface coastal water (Cai
et al., 2011), and soil heavy metal pollutions from mines (Li et al., 2014). Through the ways
like radiative transfer, geobiochemical cycle, and atmospheric circulation, the eco-effects may
interrupt the balance of Earth processes and impact terrestrial ecosystems. This is exemplified
by the case that the increase of anthropogenic aerosols in atmosphere resulted in the increase
of extreme weather phenomena, such as haze, fog, and smog, and the decrease of visibility
and sunshine durations (Kaiser and Qian, 2002; Che et al., 2007; Zheng et al., 2008).



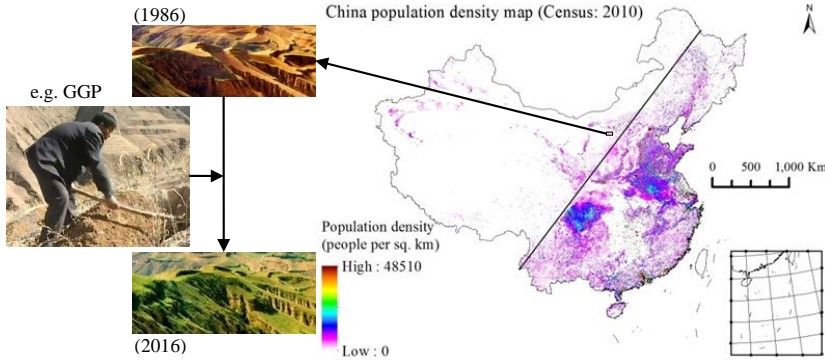



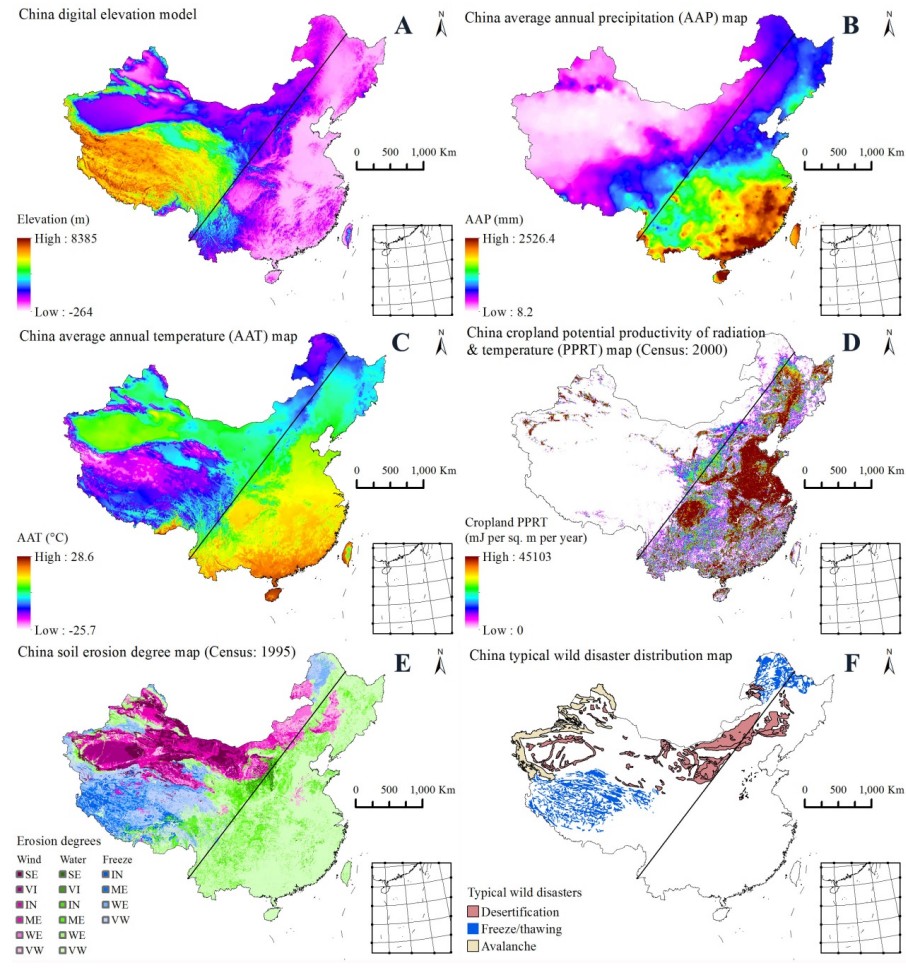



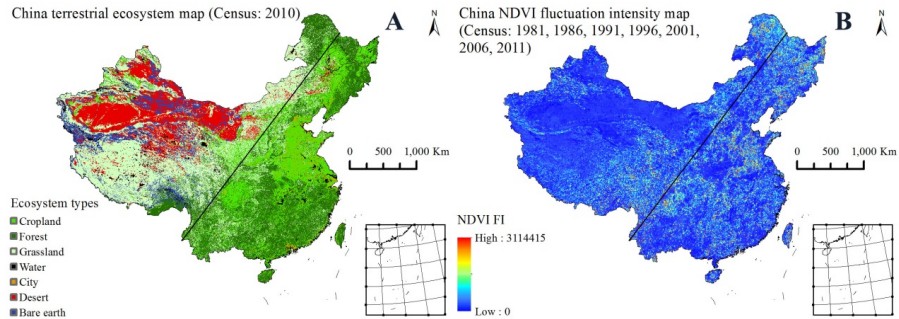



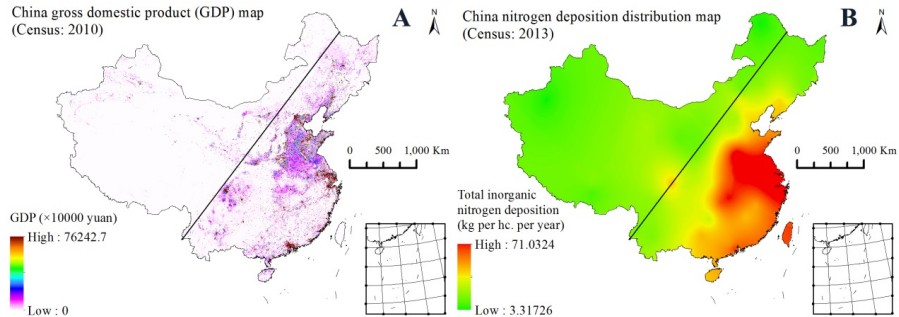



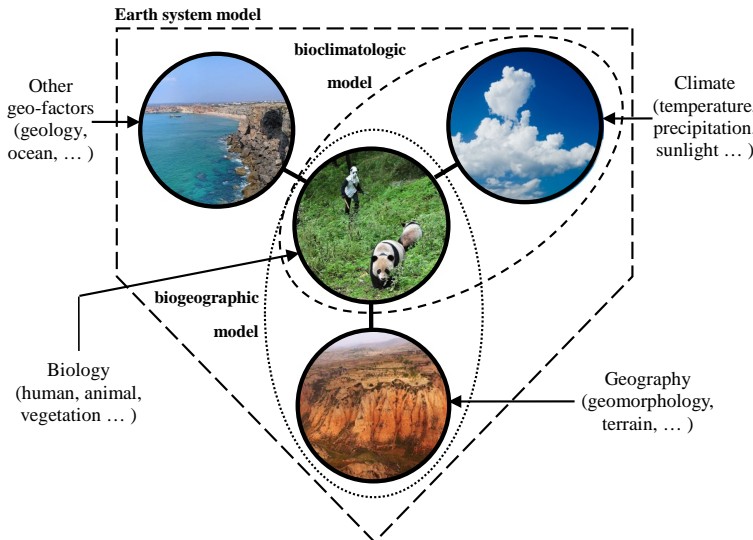