# Peer review of "Anthropogenically breaking macro-ecospatial 'chains'? case review of HU Line"

_Biogeosciences, 2019_

## Referee Comment (RC1) · Anonymous Referee #1 · 11 Dec 2019

The authors state that they used meta-analysis and remote sensing data but there is no section describing how they did that. I did not see any results compatible with the meta-analysis approach. English needs some improvement (there are typos).

---

## Author Comment (AC1) · 12 Dec 2019

Thanks for the careful comments. Actually, the authors have made the related revisions during the initial corrections before the manuscript was posted for interactive discussions. "meta-analysis", indeed, was a careless and hasty delineation in the draft, without citations either. This carelessness has been corrected by substituting it with the right approach "integrative data analysis", with the explanation and references added. For the second problem – 'English needs some improvement (there are typos)', the manuscript has been carefully checked in English use (with typos and grammatical errors revised), assisted by the GrammarCheck tool on the internet.

---

## Referee Comment (RC2) · Anonymous Referee #2 · 22 Dec 2019

General comments: Very interesting paper synthesising knowledge about the HU line and how people have tried to deal with it or break it in the past and predict its changes in the future using models. I have no major concerns about the paper, just a few comments. I think the abstract should be modified slightly in order to improve clarity for non-expert readers. I had some troubles understanding it. I would suggest working a bit on that. The concept of the H-U line is probably not enough widely known so that all readers understand what you mean when you refer to it. Just slight modifications on the text can solve this issue. I'm not sure if I'm checking a previous version of the paper, but I can still see a few typos and, I think, some mistakes in the use of language (see some of them below). Please, check the text thoroughly. I see you have

now changed to "integrative data analysis". I'm not entirely comfortable with that term because, unless there's a clear definition somewhere that I'm not aware of, it seems to be a bit broad (you can integrate plenty of different types of data in a data analysis). I suggest elaborating a bit more on the type of analysis you're actually performing. Try to answer the question of what you are integrating in your analysis. To me it just looks like a review paper using data already available somewhere else.

Minor comments: L. 20: change "human" for "humans" L. 37: "extensive interest is whether people can better the macroecosystem-related ecological" change better for improve? L. 43: Try to define or explain what you mean by "chains" the first time you mention them in the text. After reading the paper a few times, I think I now understand what it means, but it wasn't obvious to me the first time I saw it. This term also contributes a bit towards making the abstract a bit confusing. L. 82-84: That's a very interesting. In my mind, that relates to some sort of geoengineering, at least at a regional scale, but the term does not appear at all in the text. Is it just that these kinds of actions to break macro-ecospatial chains are not actually geoengineering? Consider adding something about it in the text, it may solve someone else's questions as well. L. 90: Precipitation richness? I've never seen that term before. I suggest changing that for spatially heterogeneous precipitation patterns or something similar. L. 93: "some ecosystems are preferred by life": This sentence strikes me as a bit awkward. An ecosystem has to be "alive" by definition (without living beings you would just have an abiotic habitat, not an ecosystem), so stating that some ecosystems are preferred by life I think it does not really make much sense. I suggest rephrasing the sentence towards something like: "some habitats are able to sustain more organisms, or more biomass... than others". L. 94: what do you mean with transfers? I suggest rephrasing that sentence. L. 118: "rich rainfalls": I suggest changing that for large amounts of precipitation or similar. L. 131: "thru": typo L. 195-196: Indeed! L. 259-268: So, the western side has increased its population more than the right, is that correct? I think the authors infer that the change has been, at least, partially caused by GGP-kind programs. Could you elaborate a bit more on that? If conditions are so much worse

in the west than in the east sounds counter-intuitive that population growth has been larger there in spite of GGP-programs. L. 289-303: Here the authors seem to imply that demographic changes occur mainly because of changes in regional climate and that deforestation strengthen the HU line. I think it's possible climate changes drove these changes, but what about historical events and successive governments with different agendas? Can you provide some information about geopolitics of the region for that period of time? They may also be relevant in order to understand why the HU line emerged during this period. No remarkable comments thereafter.
* * *

---

## Author Comment (AC2) · 1 Jan 2020

Response to reviewer comments Response: Sincere thanks to the reviewer's constructive comments.

General comments: Very interesting paper synthesising knowledge about the HU line and how people have tried to deal with it or break it in the past and predict its changes in the future using models. I have no major concerns about the paper, just a few comments. I think the abstract should be modified slightly in order to improve clarity for non-expert readers. I had some troubles understanding it. I would suggest working a bit on that. The concept of the HU- line is probably not enough widely known so that

all readers understand what you mean when you refer to it. Just slight modifications on the text can solve this issue. I'm not sure if I'm checking a previous version of the paper, but I can still see a few typos and, I think, some mistakes in the use of language (see some of them below). Please, check the text thoroughly. I see you have now changed to "integrative data analysis". I'm not entirely comfortable with that term because, unless there's a clear definition somewhere that I'm not aware of, it seems to be a bit broad (you can integrate plenty of different types of data in a data analysis). I suggest elaborating a bit more on the type of analysis you're actually performing. Try to answer the question of what you are integrating in your analysis. To me it just looks like a review paper using data already available somewhere else. Response: The abstract has been revised accordingly. HU Line should not confuse the readers, as its emergence just followed its definition "transition zone". The manuscript has been fully revised again, particularly aiming at typos and mistakes in English uses. Pls. refer to the blue-colored words and sentences throughout the manuscript. The method was revised as "the strategy of integrative data analysis" – the schematic framework used in this review work for regularly using the data already available somewhere else, as the reviewer said. Minor comments: L. 20: change "human" for "humans" Response: Revised as "humans" accordingly. L. 37: "extensive interest is whether people can better the macroecosystem-related ecological" change better for improve? Response: Revised as "improve" accordingly. L. 43: Try to define or explain what you mean by "chains" the first time you mention them in the text. After reading the paper a few times, I think I now understand what it means, but it wasn't obvious to me the first time I saw it. This term also contributes a bit towards making the abstract a bit confusing. Response: Explanations have been added to clarify "chains", and pls. refer to L. 44-45. L. 82-84: That's a very interesting. In my mind, that relates to some sort of geoengineering, at least at a regional scale, but the term does not appear at all in the text. Is it just that these kinds of actions to break macro-ecospatial chains are not actually geoengineering? Consider adding something about it in the text, it may solve someone else's questions as well. Response: 'geoengineering' was not involved here,

but the authors followingly added the reference to this initially climate-oriented concept in sub-Section 4.2 (pls. refer to L. 462). Thanks for the reviewer's professional advice. L. 90: Precipitation richness? I've never seen that term before. I suggest changing that for spatially heterogeneous precipitation patterns or something similar. Response: Revised as "patterns" accordingly. L. 93: "some ecosystems are preferred by life": This sentence strikes me as a bit awkward. An ecosystem has to be "alive" by definition (without living beings you would just have an abiotic habitat, not an ecosystem), so stating that some ecosystems are preferred by life I think it does not really make much sense. I suggest rephrasing the sentence towards something like: "some habitats are able to sustain more organisms, or more biomass: : : than others". Response: The sentence was maintained, since the narration here was aimed at the general-sense circumstances as highlighted in the reference work. L. 94: what do you mean with transfers? I suggest rephrasing that sentence. Response: Revised as "spatial transfers" for a more explicit definition. L. 118: "rich rainfalls": I suggest changing that for large amounts of precipitation or similar. Response: Revised as "large amount of rain" accordingly. L. 131: "thru": typo Response: Revised as "via". L. 195-196: Indeed! Response: Thanks for the reviewer's confirmation. L. 259-268: So, the western side has increased its population more than the right, is that correct? I think the authors infer that the change has been, at least, partially caused by GGP-kind programs. Could you elaborate a bit more on that? If conditions are so much worse in the west than in the east sounds counter-intuitive that population growth has been larger there in spite of GGP-programs. Response: The western side has increased its population more than the right in terms of population percentage but not in terms of population number, because of the large base of population and its change. The population increase in the western side is directly due to the economic development, instead of the GGP-kind programs. The focus here is that the population increasing in the western side leads to two kinds of counter eco-effects, as explained in the following sentences. This narration is to explain the complexity of anthropogenic eco-effects, still far from deriving an inference as the reviewer thought or giving a confirming answer to any questions

aimed at in this study. L. 289-303: Here the authors seem to imply that demographic changes occur mainly because of changes in regional climate and that deforestation strengthen the HU line. I think it's possible climate changes drove these changes, but what about historical events and successive governments with different agendas? Can you provide some information about geopolitics of the region for that period of time? They may also be relevant in order to understand why the HU line emerged during this period. No remarkable comments thereafter. Response: Yes. Climate changes and deforestation did briefly lead to demographic transfers and the formation of HU Line for such a large region. Historical events and geopolitics played minor roles during this period, as little mentioned in the overview work (Wang, 1995).

---

## Referee Comment (RC3) · Anonymous Referee #3 · 12 Feb 2020

General Comments: In the contributed manuscript, the authors reviewed literature for the question of whether anthropogenic activities can break the major macro-ecospatial transition zone in China, i.e., the Hu Huan-Yong Line. The authors discussed the natural formation and spatiotemporal stability of the line, suggesting small-scale modifiability of the line. Integrative approaches have been proposed to account for external, internal, and integral processes of future evolution of the line. In general, I find the manuscript not crystal clear and have a few major concerns. First, I missed the justification of the scientific question to be addressed here. Why were the authors motivated to explore whether the Hu Huan-Yong Line can be anthropogenically reshaped? The authors claimed that the intentionally-positive anthropogenic feedback to environment

at the microecosystem scale and how such ecological effects work have been less studied, and that China's conspicuous macro-ecospatial changes and massive environmental restorations make the line as the optimal case for filling the gap mentioned. However, around the world there are many ecological restoration projects covering broad geographical extents with clear biogeographical zoning. For example, rewilding projects across Europe promoting the return of large-bodied mammals have modified the previous boundary between humans and wilderness (Chapron et al. 2014 Recovery of large carnivores in Europe's modern human-dominated landscapes). In other words, the Introduction has not been successful in reviewing recent scientific progress on the topic, unable to identify current knowledge gap. Importantly, proposals by the authors have not been sufficiently justified. In each section of "Comprehensive exploration of HU Line evolution", the authors repeatedly argued that incorporating biogeographical, bioclimatic, and Earth system models will help predict the evolution of HU Line. However, the authors simply reviewed some cases of existing models, without elaborating on their links to HU Line and how they can be used as examples to forecast future changes in HU Line. I was expecting more discussion in depth on these key questions. Otherwise, the current status of review seems a simplistic compilation of models published, with general, unconvincing proposals. Figure 5 is not self-evident enough, as I was not informed about how these models are related to Hu Line so as to improve our ability of understanding the dynamics of HU Line. Besides, there are incorrect wording and references in the manuscript that have compromised the readability of the manuscript. For example, "undermining" (line 39) and "exploitation" (line 42) should not be the correct words. The reference in line 54 and line 204 seems irrelevant, as I didn't see that the points made by the authors have been articulated in the cited article, which emphasizes the hierarchical links among local, landscape, and macroecological processes in ecosystems at large scales. Personally, I find the writing difficult to follow. Please check the language and citations throughout the manuscript.

Specific Comments Lines 10-13. The terms "feedback" and "chain" at the beginning of the Abstract would be difficult for readers to get their exact meanings without clear

definition on the first use. Please clarify. Lines 24-26. This statement is too general. The readers would be likely wondering about more details. Please be more specific about the "fundamental implications". Lines 30-38. As mentioned earlier, please re-identify on the knowledge gap. Meanwhile, I was not convinced by the logic of linking the gap to the topic of "microecosystem-related ecological spatial (macro-ecospatial) layouts. With so many terms lumping together, please provide more clear information on each term. Also, please clarify how this topic is related to the concept of "transition" below. Lines 43-47. Why do regional- to continental-scale terrestrial transition zones resemble chains? This analogy is still unclear to me. Line 54-56. Would these two statements be too strong? It would be hard to imagine that there are few massive conservation and restoration projects in the regions mentioned. Or, please clarify the definition of "massive human improvement measures". Line 70. Has China completed its industrialization? If so, when? Lines 77-79. There are other large-scale ecological restoration projects around the world. The word "optimal" would be too strong. Lines 82-83. How this question is related to the previous texts? Line 238. Please replace "by" with "of". Lines 238-239. How general is the conclusion that HU Line would be stable at the decadal scale? Historically, the boundary between farmers and nomads, which is closely related to HU Line, can be strongly affected by precipitation over decades (Bai & Kung 2011 Climate shocks and Sino-nomadic conflict). Lines 286-288. Why are local changes justified as evidence of breaking HU Line? Would the broad-scale pattern still persist even with local-scale noises? Lines 288-303. As far as I know, the statements here on human population dynamics and climate change in China are not consistent with historical evidence. Please review the most recent advances (e.g., Wilkinson 2018 Chinese History: A New Manual; Li et al. 2018 Reconstruction of the cropland cover changes in eastern China between the 10th century and 13th century using historical documents; Ge et al. 2016 Recent advances on reconstruction of climate and extreme events in China for the past 2000 years; Chen & Kung 2016 Of maize and men: the effect of a New World crop on population and economic growth in China) and rephrase this section. Lines 312-313. Please elaborate on this point, i.e., uncertainties of the

final effects. Line 331-332. Too general. There are always uncertainties in future ecological outcomes. Please specify in details. Line 364. Throughout this section, I didn't see texts on the future evolution of HU Line. Please see the general comments above. Line 366. The world "handle" would be a little too strong. Maybe "account for"? Line 504. I didn't get the meaning of the word "integrally". Please clarify. Line 528-529. This statement is too general. Please be more specific and relevant. Line 534. What was "even though" meant here? Lines 536-537. Research on anthropogenic eco-effect on nature and its macro-ecospatial pattern is neither novel nor trivial. Please reconsider the implication of the review.

---

## Author Comment (AC3) · 21 Feb 2020

Response to reviewer comments

Sincere thanks to the reviewer's constructive comments.

Comments:

General Comments

In the contributed manuscript, the authors reviewed literature for the question of whether anthropogenic activities can break the major macro-ecospatial transition zone in China, i.e., the Hu Huan-Yong Line. The authors discussed the natural formation and spatiotemporal stability of the line, suggesting small-scale modifiability of the line. Integrative approaches have been proposed to account for external, internal, and integral processes of future evolution of the line.

In general, I find the manuscript not crystal clear and have a few major concerns. First, I missed the justification of the scientific question to be addressed here. Why were the authors motivated to explore whether the Hu Huan-Yong Line can be anthropogenically reshaped? The authors claimed that the intentionally-positive anthropogenic feedback to environment at the macroecosystem scale and how such ecological effects work have been less studied, and that China's conspicuous macro-ecospatial changes and massive environmental restorations make the line as the optimal case for filling the gap mentioned. However, around the world there are many ecological restoration projects covering broad geographical extents with clear biogeographical zoning. For example, rewilding projects across Europe promoting the return of large-bodied mammals have modified the previous boundary between humans and wilderness (Chapron et al. 2014 Recovery of large carnivores in Europe's modern human-dominated landscapes). In other words, the Introduction has not been successful in reviewing recent scientific progress on the topic, unable to identify current knowledge gap.

Thank the reviewer for mentioning the reference. The authors cited this work but used it as a comparing case for explaining the topic regarded in this study (Line 59-61). This study focused on the macro-ecospatial transition zones, instead of the landscapes as explored in the reference work. In fact, a large number of natural reserves and other measures have also been set for rewilding in China, indeed with some kinds of positive effects, but their effects are far from enough for achieving the aimed goal of anthropogenically shifting the continental-scale 'chains'.

Importantly, proposals by the authors have not been sufficiently justified. In each section of "Comprehensive exploration of HU Line evolution", the authors repeatedly argued that incorporating biogeographical, bioclimatic, and Earth system models will help predict the evolution of HU Line. However, the authors simply reviewed some cases of existing models, without elaborating on their links to HU Line and how they can be used as examples to forecast future changes in HU Line. I was expecting more discussion in depth on these key questions. Otherwise, the current status of review seems a simplistic compilation of models published, with general, unconvincing proposals. Figure 5 is not self-evident enough, as I was not informed about how these models are related to Hu Line so as to improve our ability of understanding the dynamics of HU Line.

This section is not a simplistic compilation of models, instead, a scientific proposal of the available models after their mechanism and factors appropriate for the simulation-based projection task were carefully examined. After all, almost no models have been specially developed or operated for simulation of HU Line development. To avoid the puzzle possibly left to the readers, the caption of this section has been revised as "Next: Modelling projection of HU Line evolution". This explanation is also available for Figure 5. In a sense, this section is the authors' real contribution - a scientific proposal for the future, more than a review of the previous studies.

Besides, there are incorrect wording and references in the manuscript that have compromised the readability of the manuscript. For example, "undermining" (line 39) and "exploitation" (line 42) should not be the correct words. The reference in line 54 and line 204 seems irrelevant, as I didn't see that the points made by the authors have been articulated in the cited article, which emphasizes the hierarchical links among local, landscape, and macroecological processes in ecosystems at large scales. Personally, I find the writing difficult to follow. Please check the language and citations throughout the manuscript.

"undermining" (line 39) has been revised as "investigating" and "exploitation" (line 42) has been revised as "handling". The reference in line 54 has been revised as "(e.g., Halffter et al., 2019)" (Line 57), and the reference in line 204 has been revised as "(Levy et al., 2014)" (Line 210), both with the new references added. Further, the manuscript has been thoroughly revised, particularly in English usage and citations. Pls. refer to the blue-colored words and sentences throughout the manuscript.

Specific Comments

Lines 10-13. The terms "feedback" and "chain" at the beginning of the Abstract would be difficult for readers to get their exact meanings without clear definition on the first use. Please clarify.

"feedback" has been revised as "effect" (Line 11), and "chain-like" has been revised as ", like the 'chains' of locking macroecological zonings"(Line 13).

Lines 24-26. This statement is too general. The readers would be likely wondering about more details. Please be more specific about the "fundamental implications".

The statement has been accordingly specified as "In all, the inferences and proposals of this review are of fundamental implications in scientific cognition for …". Pls. refer to the blue-colored parts in Line 24-26.

Lines 30-38. As mentioned earlier, please reidentify on the knowledge gap. Meanwhile, I was not convinced by the logic of linking the gap to the topic of "macroecosystem-related ecological spatial (macro-ecospatial) layouts. With so many terms lumping together, please provide more clear information on each term. Also, please clarify how this topic is related to the concept of "transition" below.

The authors paid a lot attention to this point in the beginning, and the logic was carefully retained, e.g., the first-time mentioning of macroecosystem followed by its reference and the lumping of "spatial" and "ecological" had very definite meanings. This should not confuse the

readers. The coherence to "transition zone" in the next paragraph was also carefully ensured, as shown by the explanation (Line 46-53).

Lines 43-47. Why do regional- to continental-scale terrestrial transition zones resemble chains? This analogy is still unclear to me.

The explanation about this analogy has already been listed in details (Line 46-53). The authors have repeatedly examined the logic of introducing the topic.

Line 54-56. Would these two statements be too strong? It would be hard to imagine that there are few massive conservation and restoration projects in the regions mentioned. Or, please clarify the definition of "massive human improvement measures".

The statement has been accordingly revised as "However, for these transition zones few massive human improvement measures such as large-scale reforestation have been effectively undertaken", with the measures clarified by listing "such as". Though some official measures have been announced for these zones, no effective effects have shown. So, the statements here were not too strong.

Line 70. Has China completed its industrialization? If so, when?

The description has been revised as "Since the beginning of industrialization in China" (Line 75).

Lines 77-79. There are other large-scale ecological restoration projects around the world. The word "optimal" would be too strong.

"the optimal" has been revised as "a preferred" accordingly.

Lines 82-83. How this question is related to the previous texts?

The statement has been added as "– "whether can people break through macro-ecospatial 'chains'?", as a summary of the above-mentioned gaps –". Pls. refer to the blue-colored part in Line 88.

Line 238. Please replace "by" with "of".

Revised as "of" accordingly. Pls. refer to the blue-colored word in Line 244.

Lines 238-239. How general is the conclusion that HU Line would be stable at the decadal scale? Historically, the boundary between farmers and nomads, which is closely related to HU Line, can be strongly affected by precipitation over decades (Bai & Kung 2011 Climate shocks and Sino-nomadic conflict).

This sub-Section discussed the temporal status of HU Line, primarily covering the last several decades. The finding as mentioned by the referee is not against this point. Actually, the finding (Bai & Kung, 2011) belongs to sub-Section 3.1 and can serve as an evidence of "HU Line modifiability". Thereby, this reference has been newly cited in sub-Section 3.1 (see Line 309-311). Thank the reviewer for supplementing the reference.

Lines 286-288. Why are local changes justified as evidence of breaking HU Line? Would the broad-scale pattern still persist even with local-scale noises?

The reviewer misunderstood the meaning of the authors. Here, the authors, substantially, gave an advice on how to break through HU Line, not mentioning local-scale noises and no evidence needed for justifying anything.

Lines 288-303. As far as I know, the statements here on human population dynamics and climate change in China are not consistent with historical evidence. Please review the most recent advances (e.g., Wilkinson 2018 Chinese History: A New Manual; Li et al. 2018 Reconstruction of the cropland cover changes in eastern China between the 10th century and 13th century using historical documents; Ge et al. 2016 Recent advances on reconstruction of climate and extreme events in China for the past 2000 years; Chen & Kung 2016 Of maize and men: the effect of a New World crop on population and economic growth in China) and rephrase this section.

Thank the reviewer for informing the recent advances. In fact, the authors had reviewed these works but had decided to not cite them. The reason is that compared to the mainstream process and causes of HU Line development in the history as clarified in the manuscript, their concerns were briefly on the performance of different aspects, such as agriculture and climate extremes. Substantially, their inferences cannot serve as the historical evidence, because they themselves were influenced by many other factors.

Lines 312-313. Please elaborate on this point, i.e., uncertainties of the final effects.

This point has been clarified as "their extreme effects being positive or negative …" (see Line 321).

Line 331-332. Too general. There are always uncertainties in future ecological outcomes. Please specify in details.

The details have been specified by adding "GGP-kind" and "across HU Line". Pls. refer to the blue-colored words in Line 341-343.

Line 364. Throughout this section, I didn't see texts on the future evolution of HU Line. Please see the general comments above.

The caption has been revised as "4. Next: Modelling projection of HU Line evolution", which both follows the style of the last two captions and fits the goal of this direction.

Line 366. The world "handle" would be a little too strong. Maybe "account for"?

Revised as "account for" accordingly.

Line 504. I didn't get the meaning of the word "integrally". Please clarify.

Revised as "uniformly".

Line 528-529. This statement is too general. Please be more specific and relevant.

The statement has been revised to be more specific and relevant. Pls. refer to the blue-colored Line 538-539.

Line 534. What was "even though" meant here?

Revised as "even".

Lines 536-537. Research on anthropogenic ecoeffect on nature and its macro-ecospatial pattern is neither novel nor trivial. Please reconsider the implication of the review.

Revised by adding "intentionally-positive" so as to be more specific. Pls. refer to the blue-colored Line 547.